# Sherry Wines: Worldwide Production, Chemical Composition and Screening Conception for Flor Yeasts

**Daria Avdanina * and Alexander Zghun ***

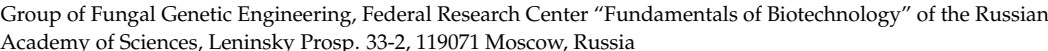

Group of Fungal Genetic Engineering, Federal Research Center "Fundamentals of Biotechnology" of the Russian Academy of Sciences, Leninsky Prosp. 33-2, 119071 Moscow, Russia

\* Correspondence: d.avdanina@gmail.com (D.A.); zzhgun@mail.ru (A.Z.); Tel.: +7-910-428-64-72 (D.A.); +7-916-974-97-69 (A.Z.)

**Abstract:** The manufacturing of sherry wines is a unique, carefully regulated process, from harvesting to quality control of the finished product, involving dynamic biological aging in a "criadera-solera" system or some other techniques. Specialized "flor" strains of the yeast *Saccharomyces cerevisiae* play the central role in the sherry manufacturing process. As a result, sherry wines have a characteristic and unique chemical composition that determines their organoleptic properties (such as color, odor, and taste) and distinguishes them from all other types of wine. The use of modern methods of genetics and biotechnology contributes to a deep understanding of the microbiology of sherry production and allows us to define a new methodology for breeding valuable flor strains. This review discusses the main sherry-producing regions and the chemical composition of sherry wines, as well as genetic, oenological, and other selective markers for flor strains that can be used for screening novel candidates that are promising for sherry production among environmental isolates.

**Keywords:** flor yeast; *Saccharomyces cerevisiae*; sherry; wine; genetic diversity; adaptation



## 1. Introduction

Sherry is a special group of fortified wines originally produced in Spain from various white grape varieties [1]. The sherry brand of wine is traditionally made from white grapes that are grown near the city of Jerez de la Frontera in Andalusia (the south of Spain) [2]. Traditional wine brands "Jerez-Xérès-Sherry" and "Manzanilla—Sanlúcar de Barrameda" from Andalusia are controlled by origin [3,4]. Young base wines can be made from grapes grown in the municipal area of nine regions—Jerez de la Frontera, Sanlúcar de Barrameda and El Puerto de Santa Maria and wine cellars in Trebujen, Chipione, Chiclana, Rota, Puerto Real and Lebrija [5,6]. However, the maturation of sherry wines until recently was limited to the so-called "sherry triangle", that is, the three cities: Jerez de la Frontera, Sanlúcar de Barrameda and El Puerto de Santa Maria [6]. For a long time, there was a confrontation between the main producers of sherry, the cities of Jerez de la Frontera and Sanlúcar, concerning mainly the technical differences between Fino and Manzanilla; as a result, in May 2021, the Consejo Regulador of the Denominations of Origin, a corporation in public law, made amendments and assumptions [7]. The new agreement states that Fino can only be produced in Jerez de la Frontera or El Puerto de Santa Maria (possibly including the new municipalities), while organic wines produced in Sanlúcar can only be called Manzanilla [7]. Additionally, an important assumption in the rules is that sherry wines can no longer be cataloged as fortified wines if they have reached a minimum alcohol content of 15% (*v/v* ethanol) without the addition of alcohol. The amendments also include the use of a number of obsolete local pre-phylloxera grape varieties for sherry production, notably the archaic varieties Mantúo Castellano, Mantúo de Pilas, Vejeriego, Perruno, Cañocazo and Beba [7,8]. These and other concessions and assumptions will allow winemakers to prepare for the future by innovating in the good, old tradition of sherry manufacturing.

The specific organoleptic properties of sherry wines are acquired as a result of a combination of several factors, the most important of which are: (i) the application of the Solera-Criadera system during the fermentation of wine material in oak barrels [9] and (ii) using the specialized "sherry" (or flor) strains of yeast, *Saccharomyces cerevisiae* [10–12]. A unique feature of sherry strains of *S. cerevisiae*, in comparison with other fermentation strains, is their ability to form a biofilm (or flor) on the surface of alcoholic wine material, in which ethanol is oxidized to acetaldehyde under the action of alcohol dehydrogenase [13–15]. The ability of sherry yeast to work under conditions of high alcohol content is their adaptive mechanism, which affects the oenological characteristics [16,17]. There are numerous works devoted to both the mechanism of flotation of sherry yeast strains and the technology of aging sherry wines [9,10,13].

Some of these works are devoted to the technology of aging sherry wines, while others consider a fundamental approach to the mechanism of flotation of sherry yeast strains. However, the molecular basis of the origin of sherry yeast strains, as well as the driving forces leading to their distribution, are still not fully understood. The use of modern achievements in molecular biology and population genetics allows us to consider existing issues in the microbiology of winemaking at a deeper level. Such knowledge is important for both initial screening and further selection of valuable yeast varieties used in the wine industry.

This mini-review is devoted to flor yeasts, with a focus on their ecological fitness and specific aspects of their genetic and metabolic activity, which can serve as selective markers for the selection of candidate strains promising for the production of sherry-type wines.

## 2. Worldwide Production of Sherry and Sherry-like Wines

In addition to Spain, Denominación de Origen (DO) sherry wines have been registered in the European Union in Italy (Sardinia), France (Jura), and Hungary (Tokay Hegyalja) under the brands Vernaccia di Oristano [18], Vin Jaune [19], and Szamorodni [20], respectively (Figure 1, Table 1). To make these, winemakers use local white grape varieties *Vernaccia* and *Savagnin* in Italy and France, as well as *Furmint*, *Harslevelű*, and *Sárga Muskotály* varieties in Hungary. DO sherry wines are produced according to the classical flor technology with biofilm formation [21].

**Table 1.** Major sherry and sherry-like wine production worldwide and their analytical characteristics.

| Country | Region/City | Brand | Alcohol, % | Sugar, g/L | Type | Grape Varieties | References |
|---|---|---|---|---|---|---|---|
| Spain | Andalusia (Jerez de la Frontera, El Puerto de Santa María, Sanlúcar de Barrameda) | Fino<br>Manzanilla<br>Amontillado<br>Oloroso<br>Palo Cortado | 15<br>15<br>16–22<br>17–22<br>17–22 | 0–5 | Dry (Vinos Generosos) | Palomino | [6,22] |
| | | Dry<br>Medium<br>Pale Cream<br>Cream | 15–22<br>15.5–22<br>15.5–22<br>15.5–22 | 5–45<br>5–115<br>45–115<br>115–140 | Sweet (Vinos Generosos de Licor) | Mix of Palomino, Pedro Ximénez, Moscatel de Alejandria | |
| | | Moscatel | 15–22 | 160+ | Natural Sweet (Vinos Dulces Naturales) | Moscatel de Alejandria | |
| | | Dulce<br>Pedro Ximénez | 15–22<br>15–22 | 160+<br>212+ | | Pedro Ximénez | |
| Italy | Sardinia | Vernaccia di Oristano Bianco | 15 | N/D [1] | Dry | Vernaccia di Oristano | [18,22] |
| | | Vernaccia di Oristano Superiore | 15.5+ | | | | |
| | | Vernaccia di Oristano Riserva | 15.5+ | | | | |
| | | Vernaccia di Oristano Liquoroso | 16.5+ | | Sweet | | |
| France | Jura | Vin Jaune | 14 | 0.4 | Dry | Savagnin | [19] |

**Table 1.** *Cont.*

| Country | Region/City | Brand | Alcohol, % | Sugar, g/L | Type | Grape Varieties | References |
|---|---|---|---|---|---|---|---|
| Hungary | Tokaj-Hegyalja | Tokaji Szamorodni | 13.5 <br> 13.5 | 9 <br> 45+ | Dry <br> Sweet | Furmint, Hárslevelű, and Sárga Muskotály | [20] |
| USA | California | Sheffield Very Dry Sherry | 17 | 0.4–1.4 | Dry | Local Grape Varieties | [10,23] |
| | | Sheffield Cellars Cream Sherry | 18 | 10.6 | Medium Dry | | |
| | | Paul Masson Pale Dry Sherry | N/D | N/D | Dry | | |
| | | Paul Masson Medium Dry Sherry | | | Medium Dry | | |
| | | Golden Cream Sherry | | | Sweet | | |
| Australia | McLaren Vale | Tinlins Dry Sherry-Apera | 17.5 | N/D | Dry | Local Grape Varieties | [10,23] |
| | | Tinlins Cream Sherry-Apera | | | Sweet | | |
| | | Tinlins Sweet Sherry-Apera | | | Sweet | | |
| | | Tinlins Muscat Sherry-Apera | N/D | | Sweet | | |
| Africa | South Africa | Old Brown Sherry Cream Sherry | N/D | N/D | Dry <br> Medium Sweet | Local Grape Varieties | [10,23] |
| Cyprus | Cyprus | Emva Cream Sweet Sherry | 15 | N/D | Sweet | Local Grape Varieties | [23] |
| | | Emva Medium Dry Sherry | 17.5 | | Medium Dry | | |
| UK | Bristol | Harveys Fino | 15 | 0 | Dry | Palomino | [24] |
| | | Harveys Fine Old Amontillado | 19 | 5 | | | |
| | | Harveys Palo Cortado | 19 | 19 | | | |
| | | Harveys Medium Dry | 17.5 | 38 | Medium Sweet | Palomino, Pedro Ximénez | |
| | | Harveys Rich Old Olorosso | 20 | 53 | Sweet | | |
| | | Signature by Harveys | 19 | 120 | Sweet | | |
| | | Harveys Bristol Cream | 17.5 | 135 | Sweet | | |
| | | Harveys Pedro Ximénez | 16 | 440 | Sweet | Pedro Ximénez | |
| Russia | Krasnodar Krai (stanitsa Golubitskaya) | Xeres Tamanskii | 20 | 30 | Medium Sweet | Sauvignon Blanc | [25] |
| | Rostov-on-Don | Xeres Krepki Donskoi | 19 | 3 | Dry | Aligote, Plavai, and Riesling | [26] |
| | Republic of Dagestan (Derbent) | Xeres Dagestanskii | 19 | 30 | Medium Sweet | Rkatsiteli, Narma, Gulyabi, Khatmi, Ag-izym | [27] |
| Republic of Crimea | Yalta | Xeres Oreanda | 16 | 0 | Dry | Aligote, Albillo, Kokur | [28] |
| | | Xeres Massandra | | 25 | | Sersial, Verdelho, Albillo | [29] |
| | | Xeres Magarach | 19.5 | | Medium Sweet | Aligote, Sauvignon, Rkatsiteli | [30] |
| Ukraine | Odessa | Shabo Reserve Sherry Dessert | N/D | N/D | Medium Sweet | Muscat Ottonel | [23,31] |

**Table 1.** *Cont.*

| Country | Region/City | Brand | Alcohol, % | Sugar, g/L | Type | Grape Varieties | References |
|---|---|---|---|---|---|---|---|
| Moldova | Ialoveni | Ialoveni Sec Reserva | 16 | 0–5 | Dry | Muscat de Ialoveni, Aligote, Traminer, White Pinot, Rkatsiteli | [32] |
| | | Ialoveni Armonios Reserva | 18 | 15 | Medium Dry | | |
| | | Ialoveni Tare Reserva | 20 | 30 | Medium Sweet | | |
| | | Ialoveni Desert Reserva | 19 | 90 | Sweet | | |
| Armenia | Ashtarak, Shaumyan, Echmiadzin | Ashtarak | 20 | 30 | Medium Sweet | Voskeat, Chilar | [33] |

[1] N/D—No data.

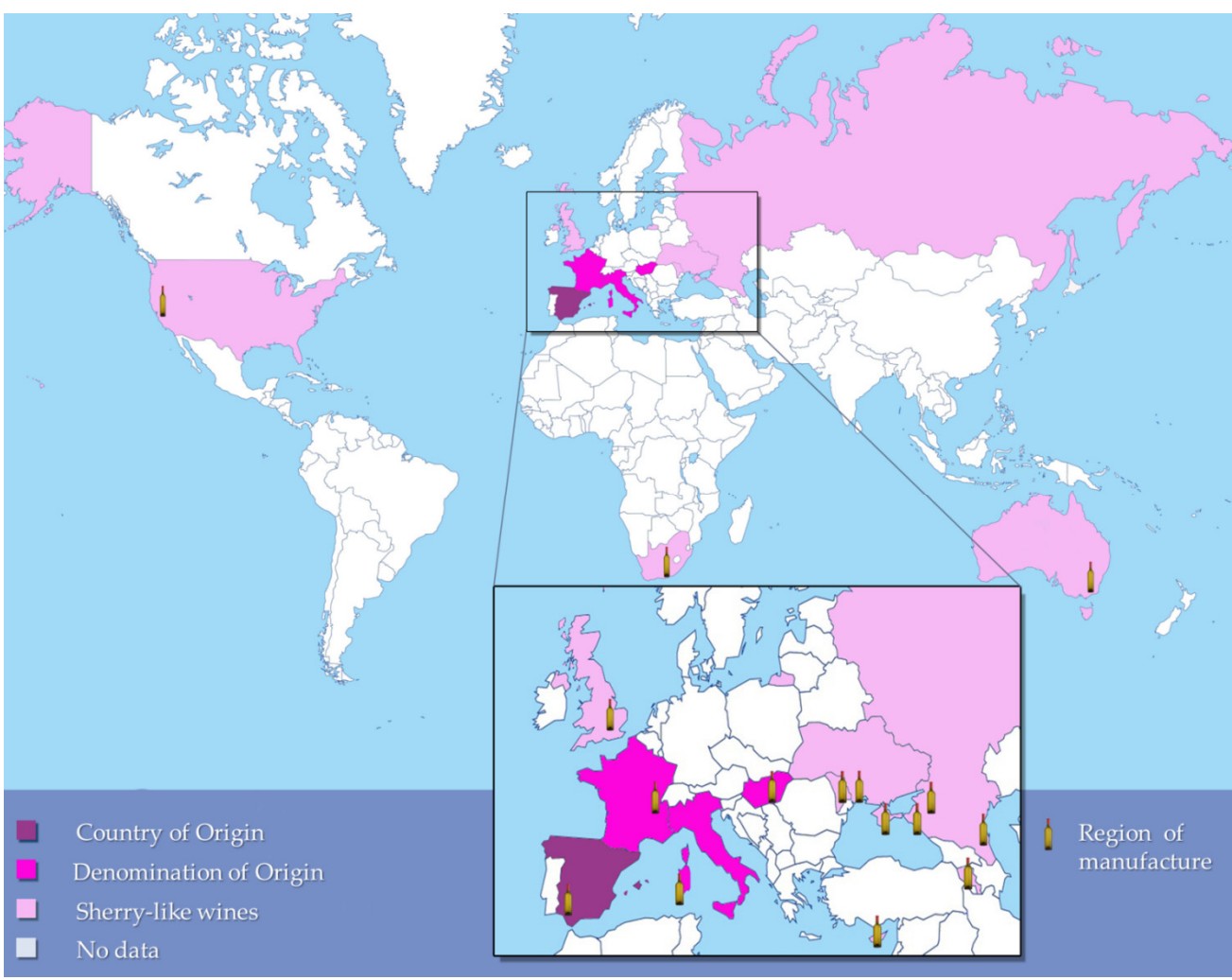

**Figure 1.** Worldwide production of sherry and sherry-like wines. Spain, the origin of sherry wines—colored in dark purple; countries with registered Denominación de Origen (DO) production of sherry wines—colored in fuchsia; countries that produce sherry-like wines—colored in light pink; countries for which there are no data on the production of sherry-like wines—colored in white. The bottle icon indicates the region in which the sherry or sherry-like wine is produced.

A number of other countries use flor technology as well as submerged fermentation and sherry wine aging technology for sherry-like wine (SLW) production (Figure 1, Table 1). Australia, Southern Africa, and Cyprus produce SLWs of different kinds, both in barrels and using New World winemaking techniques [23]. In the USA (California), a huge range

of SLWs are produced, from dry to liqueurs and sweet, both by the film-forming method with flor yeasts and by the "backing" method in steel tanks with a special temperature regime, without yeast [8,23]. Popular in the UK, the SLW Harveys is made from imported must from traditional Spanish grape varieties [24]. A variety of SLWs with the common name Ialoveni are produced in Moldova, using the grape varieties *Muscat de Ialoveni*, *Aligote*, *Traminer*, *White Pinot*, and *Rkatsiteli* [32,34–36]. In Ukraine, in the Odessa region, SLW is produced under the name Shabo using the *Muscat Ottonel* grape variety [23,31]. A lot of SLWs are produced in Yalta, such as Xeres Oreanda (from grape varieties *Aligote*, *Albillo*, *Kokur*) [28], Xeres Massandra (from grape varieties *Sersial*, *Verdelho*, *Albillo*) [29], and Xeres Magarach Massandra (from grape varieties *Aligote*, *Sauvignon*, *Rkatsiteli*) [31]. In Russia, SLW is produced in Krasnodar Krai (under the name Xeres Tamanskii from the *Sauvignon Blanc* grape variety) [25], in Rostov-on-Don (under the name Xeres Krepki Donskoi from the *Aligote*, *Plavai*, and *Riesling* grape varieties) [26], and in the Republic of Dadestan (under the name Xeres Dagestanskii from grape varieties *Rkatsiteli*, *Narma*, *Gulyabi*, *Khatmi*, and *Ag-izym*) [27]. In Armenia, strong white vintage SLW is produced in the regions of Ashtarak, Shaumyan, and Echmiadzin from the local grape varieties *Voskeat* and *Chilar* [33].

## 3. Sherry Wine Manufacturing Process

The history of the production of Spanish sherry goes deep into the past and is an integral part of the culture of the Andalusian region, which became possible due to the mild climatic conditions (3000 h of sunshine per year, an average annual temperature of 25 °C) [3]. *Palomino Fino*, *Pedro Ximenez*, and *Moscatel* grape varieties are traditionally used to produce high-quality sherry wines. The main types of classic dry sherry wines include the so-called Vinos Generosos (brands: Fino, Oloroso, Amontillado, Palo Cortado, and Manzanilla), Table 1. These dry wines are made from the same base wine but subjected to different aging procedures [38]. There are also blended fortified liqueur sweet wines in the category Vinos Generosos de Licor (brands: Dry, Pale Cream, Medium and Cream) and natural sweet wines in the category Vinos Dulces Naturales (brands: Pedro Ximénez, Moscatel and Dulce).

### 3.1. Production of Fino Sherry Wines

In Spain, the process of making sherry wines of various types has a strict gradation by stages (Table 2, Figure 2). Fino brand wines are biologically aged dry sherries and go through successive stages of preparation. For Fino, first-pressed *Palomino* grapes must enter the stage of alcoholic fermentation in order to obtain a young wine (*sobretablas* stage, Spanish). At this stage, in addition to *S. cerevisiae*, it is possible to detect other wine microorganisms (Figure 2), such as lactic acid bacteria and non-*Saccharomyces* yeasts, whose growth may lead to organoleptic deviations and deterioration of the wine [39]. To prevent lactic acid bacteria development, lysozyme could be used, as its muramidase activity has an antibacterial action [39]. Concerning non-*Saccharomyces* yeasts, these could be *Candida stellata*, *Dekkera anomala*, *Hanseniaspora guilliermondii*, *Hanseniaspora uvarum*, *Issatchenkia terricola*, *Starmerella bacillaris*, *Lachancea thermotolerans*, and *Torulaspora delbrueckii*, which are often found in nature on the surface of grapes [40–44]. Further, the wine material is subjected to fortification (*encabezado* stage, Spanish) by adding ethanol at 15.0–15.5% (*v/v* ethanol), while the composition of the microflora changes dramatically—ethanol-sensitive microorganisms are almost completely replaced by sherry strains of *S. cerevisiae*, a distinctive feature of which is the ability to form a biofilm on the surface of the wine material—flor 1–3 cm thick [45]. Additionally, an important role in this process is played by such adaptive advantages of *Saccharomycetes* as high fermentation rate, resistance to alcohol, ability of anaerobic growth and rapid switching of metabolism from enzymatic to oxidative [42,46,47]. This sherry stage takes place in the criadera-solera system in oak barrels with a volume of 500–600 L [5,40]. It was shown recently that, in addition to *Saccharomycetes*, the wine microbiome at this stage may contain small amounts, of less than 0.5%, of the yeasts *W. anomalus*, *P. membranaefaciens*, *P. kudriavzevii*, and *P. manshurica* [5]. At

present, their role in the maturation of sherry wines has not been elucidated [5]. In order for the flor to grow on the surface of the wine, the barrels are filled to 4/5 of their volume and arranged in pyramidal rows called "criaderas" [1,5]. There may be several such tiers, which are numbered from bottom to top, for example, for three tiers: first criadera, second criadera, and third criadera. The wine material is initially loaded into the uppermost tier, kept for a set time, and then enters the lower ones, until it reaches the very last, lower, "solera", containing the oldest wine in the system. From here, the wine is sent for bottling. The bottom row of barrels is emptied by no more than 40% per year. Thus, the amount of wine withdrawn from the lower tier is compensated by the same amount from the upper tier [48]. The process of wine renewal is carried out 3–4 times a year; there is a constant circulation of wine material, and its organoleptic properties remain unchanged. The biological aging of wine material during its aging under the flor largely determines the properties of Fino brand wines and lasts at least 5 years [49,50]. The distinctive fruit and floral notes are formed in taste and aroma due to due to the formation of such compounds as farnesol, β-citronellol, β-ionone, and 1,1-diethoxyethane [49]. Additional characteristics are cheesy notes, such as rancidity and pungency due to the appearance of butanoic acid and acetaldehyde, respectively [49]. Fino brand wines have been shown to contain very little or no phenolic aldehydes [51]. This brand of sherry wine is also characterized by a pale shade, which is explained by the presence of a biological aging stage. It is assumed that the flor on the surface prevents the diffusion of atmospheric oxygen into the wine material, thereby preventing the appearance of brown pigments in it [51].

**Table 2.** Major properties of dry sherry wines.

| Major Properties | Sherry Wine Type | | |
|---|---|---|---|
| | *Fino* | *Amontillado* | *Oloroso* |
| The grape variety used in the manufacture | Palomino | Palomino | Palomino |
| Soil type used for cultivating grapes | White Albariza | White Albariza | White Albariza |
| Extraction of grape must | First | First | Second |
| Type of Aging | Biological | Mixed (Biological and Oxidative) | Oxidative |
| Final alcohol concentration, % | 15–15.5 | 16–22 | 17–22 |
| Main aromatic notes | Floral, Fruity, Cheesy, Chemical, Balsamic | Nutty, Smoky, Herbs, Ethereal | Nutty, Smoky, Dry Fallen Leaves, Spices, Truffles, Ethereal |
| Color | Pale Yellow | Golden-amber to Brown | Mahogany |

### 3.2. Production of Oloroso Sherry Wines

The Oloroso brand includes dry sherry wines obtained by oxidative aging. For the production of the base wine, which is subsequently turned into a finished Oloroso, the must of the second extraction is used (Table 2, Figure 2). Unlike Fino, Oloroso sherry fortifies up to 17–22% (*v/v* ethanol) [23,38,51]. Such fortification kills the flor of *S. cerevisiae*, that is formed during the settling of young wine, and eliminates the further presence of any yeast in the wine. Oloroso is aged like Fino for a minimum of 5 years [50]. However, the older the wine, the more worthy its blending organoleptic properties become. Oloroso has distinctive notes of tobacco and ether, dry fallen leaves, spices and truffles, associated with components such as ethylguayacol, ethyl acetate, sotolon, and others [52]. The wine of this brand can eventually change its color from amber to red-brown due to the formation of quinones during the oxidation of phenolic components, the type and amount of which depends on the grape variety and production technology [38,53]. Wooden barrels allow oxygen to pass

through [54], and as a result of the absence of a yeast film, some polyphenols condense, in particular flavan-3-ol monomers with acetaldehyde [38]. Additionally, progression in browning is due to the condensation–oxidation reaction involving glyoxylic acid, which is formed by the oxidation of tartaric acid in the presence of certain metals [38]. Additionally, when Oloroso ages, a number of components are extracted from oak barrels, and its acidity, $SO_2$, Fe ions and $O_2$ content, temperature also change, which form a special organoleptic bouquet [38]. Unlike biologically aged Fino sherry, Oloroso sherry produces phenolic aldehydes of the benzoic type (vanillin and lilac aldehyde), cinnamon type (coniferyl and lilac aldehydes) and coumarins (scopoletin and esculetin) due to oxidative aging in oak barrels [51].

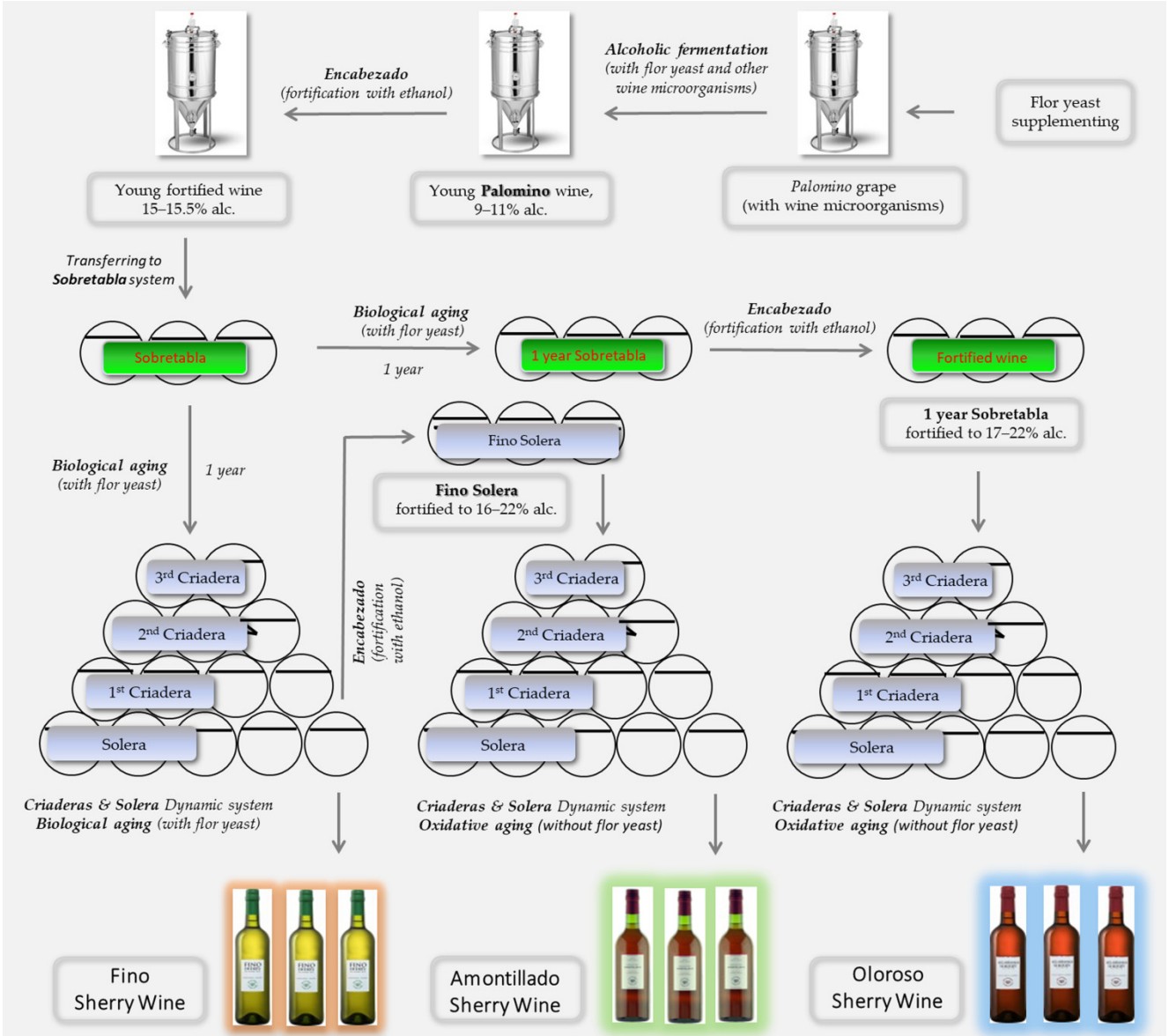

**Figure 2.** Manufacturing process for sherry wines: Fino, Amontillado, and Oloroso.

### 3.3. Production of Amontillado Sherry Wines

Amontillado is a dry sherry that has undergone a long biological aging period followed by full oxidative aging (Table 2, Figure 2). This means the Amontillado wine made from the same Palomino grape is first biologically aged like Fino type sherry, then subjected to fortification to a minimum 16.0% (*v/v* ethanol), which avoids the feasibility of *S. cerevisiae*

cell development on the surface of wine, and afterwards it is transferred into an oxidative aging system [55]. Chemometric data analysis of Fino and Amontillado has marked great differences between two sherry wines [55]. Technologically, this is the most complex sherry, the taste and color qualities of which are between Fino and Oloroso [51,56,57] and depend on many factors, including average aging time, the time it has been in contact with oak wood, etc.

## 4. Chemical Compounds Characteristic to Sherry Wines

The manufacturing of sherry wines is a unique, carefully regulated process, from harvesting to quality control of the finished product, involving dynamic biological aging in the "criadera-solera" system or some other techniques (Section 2). During this process, all the main parameters from the temperature of the cellar to the time of sherrying are strictly maintained. In addition to manufacturing technology, the composition of substances in sherry wines is influenced by: (i) the soil in which the grapes grew; (ii) the grape variety used for sherry production; (iii) the microorganisms (primarily flor yeast *S. cerevisiae*) involved in sherrying and the ethanol content; and (iv) the barrel material in which sherrying took place (Figure 3). As a result, sherry wines have a characteristic and unique chemical composition that determines their organoleptic properties (such as color, odor, taste) and distinguishes them from all other types of wine [1]. These compounds include a number of specific and distinctive carbonyls, organic acids, alcohols, volatile phenols, esters, terpenes, and lactones, as well as glycerol and nitrogen-containing compounds (Figure 3).

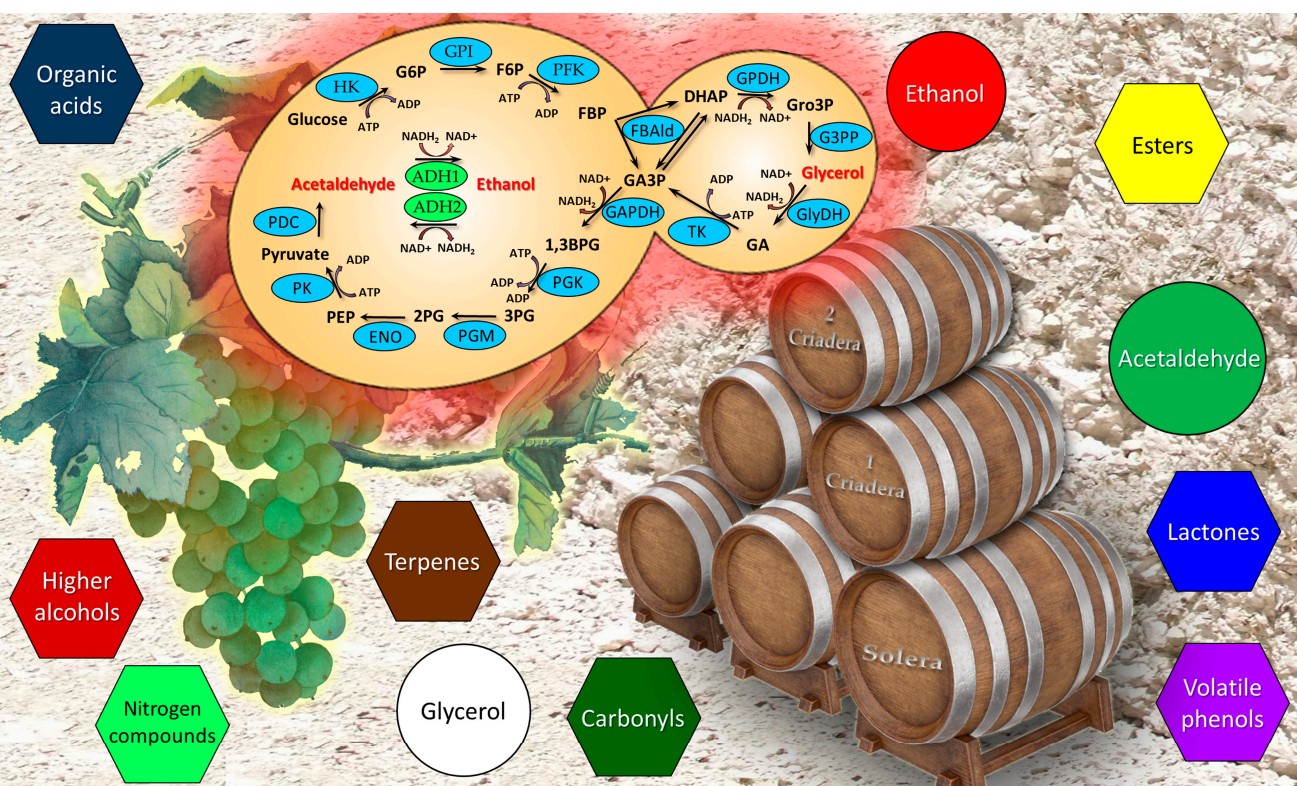

**Figure 3.** Major chemical compounds found in sherry wines. The names of a group of compounds are placed in a hexagon, whereas the names of individual compounds are placed in a circle. Fermentation reactions leading to the formation of glycerol, acetaldehyde and alcohol are conventionally placed in the mother and daughter cells of the budding *S. cerevisiae*, while all reactions occur both in the body of the mother and daughter cells. Compounds involved in metabolism: G6P—glucose 6-phosphate; F6P—fructose 6-phosphate; FBP—fructose 1,6-bisphosphate; DHAP—dihydroxyacetone phosphate; Gro3P—glycerol 3-phosphate; GA3P—glyceraldehyde 3-phosphate; GA—glyceraldehyde; 1,3BPG—1,3-bisphosphoglyceric acid; 3PG—3-phosphoglyceric acid; 2PG—2-phosphoglyceric acid; PEP—phosphoenolpyruvate. Enzymes that carry out metabolic reactions: HK—hexokinase (EC no.: 2.7.1.1); GPI—glucose-6-phosphate isomerase (EC no.: 5.3.1.9); PFK—phosphofructokinase (EC no.: 2.7.1.11); FBAld—fructose 1,6-diphosphate aldolase (EC no.: 4.1.2.13); GPDH—glycerol-3-phosphate dehydrogenase (EC no.: 1.1.1.8); G3PP—glycerol-3-phosphate phosphatase (EC no.: 3.1.3.21); GlyDH—glycerol dehydrogenase (EC no.: 1.1.1.6); TK—triokinase (glyceraldehyde 3-phosphotransferase, EC 2.7.1.28); GAPDH—glyceraldehyde 3-phosphate dehydrogenase (EC 1.2.1.12); PGK—phosphoglycerate kinase (EC no.: 2.7.2.3); PGM—phosphoglycerate mutase (EC number: 5.4.2.11); ENO—enolase (phosphopyruvate hydratase, EC no.: 4.2.1.11); PK—pyruvate kinase (EC 2.7.1.40); PDC—pyruvate decarboxylase (EC 4.1.1.1); ADH1—alcohol dehydrogenases 1 (EC 1.1.1.1); ADH2—alcohol dehydrogenases 2 (EC 1.1.1.2).

One of the main volatile compounds is ethanol, which is produced from grape sugars during yeast fermentation [58]. Ethanol in yeast is obtained for from acetaldehyde (a product of glucose metabolism) and NADH by the enzyme alcohol dehydrogenase I (ADH1) (Figure 3) [59]. The concentration of ethanol is highly dependent on the initial sugar in grape must. In the case of sherry wines, its content varies between 15–15.5% and 18–22% (*v/v* ethanol) for Fino and Oloroso wines, respectively. During biological aging, ethanol content decreases because of its consumption by flor yeasts as a source of carbon and energy [38]. The specific concentration of ethanol depends on various factors such as yeast strain, cell temperature, ratio between velum surface and wine volume, ratio between velum surface and stem and air volume in the cask [10].

One of the most important components of sherry is acetaldehyde, which is a precursor to a large number of other compounds involved in the aroma of sherry [60]. It is formed during the oxidation of ethanol under the action of alcohol dehydrogenase II (ADH2), which, unlike alcohol dehydrogenase I, oxidizes ethanol back to acetaldehyde (Figure 3) [58]. In addition to the ethereal notes of overripe apple, acetaldehyde brings the pungent aroma of Fino sherry [13]. Moreover, it is a precursor of 1,1-diethoxyethane [61], one of the main acetals in sherry wines, and is also involved in the formation of other aromatic compounds, such as acetoin and 2,3-butanodiol [58]. For the manifestation of a characteristic sherry tone, the content of acetaldehyde should be at least 350 mg/L [41]. In the process of biological aging, its level can reach 1000 mg/L, and the levels of acetals, in particular diethyl acetal, can reach 50–60 mg/L [62].

In the process of biological aging, the amount of most organic acids decreases. At the first (*sobretablas*) stage, malolactic fermentation is observed [58]. This means that malic acid is converted into lactic [63]. Then, at the *encabezado* stage, during the process of biological aging, the amount of most organic acids decreases. For example, lactic and acetic acids can be metabolized by flor yeast and converted into acetaldehyde and other by-products [38]. The content of tartaric acid decreases in the process of biological aging due to the precipitation of its crystals [58]. Gluconic acid is a marker for determining the degree of rot damage to grapes and, accordingly, the original must. It is allowed to use must with a gluconic acid concentration below 1 g/L. The level up to 1–2 g/L may indicate an initial stage of grape infection by fungi *B. cinerea*, whereas higher levels (up to 2–3 g/L might be interpreted as the result of the activity of acidobacteria [64]. At an acceptable level of gluconic acid in the must, the yeast will usually metabolize this acid without altering the quality of the final wine. The concentration of other fatty acids may increase or decrease depending on the metabolism of the yeast. Thus, an increase in the content of butanoic, isobutanoic and some other acids, as well as medium-chain fatty acids (hexanoic, octanoic and decanoic) is of the opposite nature and may completely disappear during aging [65].

Higher alcohols play an important role in the formation of the aroma of sherry wines. They provide organoleptic hues ranging from floral and fruity to musty, waxy and roasted tones [1]. It is assumed that the amount of higher wine alcohols decreases with a long process of oxidative action [50,55]. However, there are studies that show that the concentration of alcohols such as propanol, isobutanol, isoamyl and phenethyl alcohols can remain constant or increase during the biological aging of sherry wines [65].

Esters, like other volatile compounds, undergo metabolic changes as the wines mature, which is due to the activity of esterase enzymes, the growth rate of *S. cerevisiae* cells, and the formation of various compounds. For this, there are special procedures during sherrying that lead to intensification of the activity of various groups of enzymes, including esterases [50].

It is assumed that there are three different chemical pathways for the transformation of phenolic compounds during sherrying: the oxidation of phenols, condensation of flavans with glyoxylic acid, formed during the oxidation of tartaric acid, and direct condensation of phenols with acetaldehyde, as was shown for *S. cerevisiae* races *capensis* and *bayanus*, two typical flor yeasts involved in the biological aging of sherry wines [66]. The total polyphenol content in Fino wines is low [51] and decreases with time, so these wines are pale yellow in color, but they sometimes tend to be browner when bottled, as flor yeasts can retain brown pigments when exposed to oxygen [12]. During biological aging, in contrast to the oxidative process, the yeast protects the wine from atmospheric air with its veil, and traps the compounds that lead to browning. It was noted that periodic aeration in the production of Fino DO Montilla-Moriles wines more significantly affects the activity of all enzymes, including those affecting the metabolism of polyphenols, than without air [50]. Additionally, this, unfortunately, can reduce the quality of Fino. In the case of oxidative aging without flor formation because of a high level of ethanol, a progressively darker color with a mahogany tone occurs [38].

Traditionally, sherry wines are aged in wooden barrels; the types of barrels for biological and oxidative aging may differ. However, regardless of the type of material from which the barrels are made, certain lactones are extracted during the sherry process, with lactones initially present in both *cis* and *trans* isomeric forms. It has been shown that the content of the *trans* form decreases with age [59,60]. In sherry wines, special attention is paid to sotolone [67], one of the remarkable lactones formed during biological aging and increased during oxidative aging. It was first detected in the wines of Jaune (France) [68] but has also been found in sherry and some botrytised white wines [50]. Sotolone is responsible for the manifestation of nuts, curry and cotton candy notes in the aroma and taste of wine and is made from 2-ketobutyric acid and acetaldehyde [68]. Another lactone, soleron, is also typical to sherry wines but its impact on wine aroma has been shown to be very low [38].

Glycerol also affects the taste of sherry wines, giving them a peculiar sweetish aftertaste and some viscosity [62], which appears at the beginning of the fermentation process and decreases over time, since the yeast uses it as a carbon source (Figure 3) [65], and its concentration decreases during fermentation. In this regard, the concentration of glycerol can serve as an indicator of wine aging [38].

One of the key sources (for the normal functioning of flor yeast) are the sources of nitrogen intake. Normal nitrogen intake is important for the growth and vital activity of floristic yeast, which can use ammonium ions, oligopeptides, proteins, amides, and biogenic amines, as well as nucleic acids and amino acids, especially L-arginine and L-proline [69,70]. In the "oxygen-free" production of sherry, L-proline cannot be assimilated as a source of nitrogen since molecular oxygen is required for its catabolism [71]. Amino acids such as threonine, methionine, cysteine, tryptophan, and proline can serve as a redox agent to balance the redox potential under oxygen-limited conditions by oxidizing NADH in excess [58].

Aging sherry in wooden barrels is an important step in the production of wines in general, as this process gives the final product organoleptic characteristics that could not be achieved if stainless steel containers were used. Sherry is traditionally produced using American oak. However, there are investigations on the use of French and Spanish oak barrels or even chestnut casks. Nonetheless, such approaches showed satisfactory parameters in terms of oenological characteristics.

## 5. Genetic and Oenological Markers for Screening Flor Strains of *S. cerevisiae*

The main representative of yeast in sherry film is *S. cerevisiae*, whose genome was completely sequenced in 1996, for which an international consortium of scientists from several countries was created [72]. Over the past 25 years, largely as a result of the application of NGS technology, more than 200 different strains of *S. cerevisiae* have been completely sequenced [62]. This made it possible to carry out a comparative analysis of the changes occurring in the genomes of flor strains of *S. cerevisiae* [73]. It turned out that there are a number of cumulative factors that determine the unique properties of such yeast, one of which is a specific deletion of 24 bp or an insertion of "C" in the ITS1 region [74]. Flor yeast is also known to have a characteristic 111 bp deletion in the promoter of the *FLO11* gene, the most important cellular adhesin responsible for cell aggregation and film formation [69,75,76]. This deletion has been shown to be characteristic of Spanish, French, Italian, and Hungarian sherry strains [69]. The role of this deletion is to inactivate long noncoding (nc)RNAs, ICR1 [77]. The 3.2 kb ICR1 ncRNA is initiated upstream from the *FLO11* ORF (~3.4 kb) and is transcribed across much of the large *FLO11* promoter [78], repressing *FLO11* transcription in *cis* [77]. Inactivation of this negative regulator, ICR1, stimulates the synthesis of the Flo11 protein in flor yeast [77]. The sizes of *FLO11* coding sequences in yeast *S. cerevisiae* vary from 3 to 6 kb, which is mainly due to the variation in the number of repeats in the central domain. Initially, sherry strains were shown to encode longer *FLO11* variants with a higher degree of hydrophobicity [79]. However, later it was shown that the central domain of *FLO11* is extremely unstable under nonselective

conditions and that other repeats, as well as the expression level of the *FLO11* gene itself, play an important role in the ability of yeast to form flor [80].

Comparative genomics and transcriptomics are currently being used to understand the molecular basis by which a strain of *S. cerevisiae* becomes flor. [73]. The use of these approaches makes it possible to control changes in the expression of numerous genes involved in unique cell–cell adhesion, stress resistance, iron uptake, nitrogen, carbon, and lipid metabolism, and the production of aromatic compounds, which determine the specifics of sherry yeast functioning. Additionally, such a modern approach to understanding the adaptive mechanism of the yeast *S. cerevisiae* makes it possible to develop convenient genetic markers for determining strategies for the directed selection of strains suitable for sherrization.

### 5.1. Genetic Markers of Flor Yeast

One of the rapid approaches for screening natural isolates for flor yeast is based on genotyping for the ITS marker. For this purpose, restriction fragment length polymorphism (RFLP) of the amplified ITS1 region is used with selected characterizing restriction enzymes such as *Hae*III [17,41,81] or *Cfo*I and *Hin*fI [74,81]. As a result, the yeast is sorted into "wine" and "flor" strains. To divide flor yeast races into "Spanish" (deletion of 24 bp in the ITS1 region) and "French" (insertion "C" in the same fragment) [74], PCR fragments with ITS1 are treated with *Hha*I restriction enzyme [16].

However, the use of only the ITS region as a marker for determining the "flor" strain is insufficient; therefore, at present, the range of selective markers has expanded significantly [17,41,82]. Currently, the *FLO11* promoter polymorphism region is also used as a screening marker for sherry strains. For example, a primer pair is used that anneals up- and downstream of the heterogeneous region of this promoter. As a result, a 111 bp truncated product is amplified from flor strains (due to the [-1313]–[-1203] deletion) [75].

Additionally, polymorphisms of the *YDR379C-A* gene, which encodes the SDH6 subunit of mitochondrial succinate dehydrogenase [83], are effective selective markers of flor strains [17,41]. This amplifies the region, 885 bp long; polymorphism is determined after treatment with restriction enzyme *Afl*III, which has one restriction site in sherry yeast sequences (with the formation of characteristic fragments, 450 bp and 350 bp long) and no restriction sites in wine yeast sequences (when treated with this restriction enzyme, the fragment size does not change) [17]. Thus, for the primary screening of candidate sherry strains, a combination of markers ITS1+YDR379C-A, or ITS1+FLO11, or ITS1+YDR379C-A+FLO11 can be used [17].

### 5.2. Oenological Markers of Flor Yeast

Candidate strains selected at the first stage are tested under microwinemaking conditions with subsequent analysis of oenological control parameters of the finished product according to the official methodology described by the OIV in the Compendium of International Methods of Analysis of Wines and Musts [84]. When recommending strains for sherrization, first of all, their enzymatic activity, ability to form volatile acids, ability to form a biofilm, etc., should be checked [37,41]. The mechanism that regulates the formation of a yeast film on the surface of the wine material is very complex. Recent studies have shown that the transcriptional regulators Cyc8p and Tup1p in *S. cerevisiae* cells antagonistically affect its formation [85,86]. The low fermentation activity of the strains leads to an excess content of glucose in grape must, which immediately leads to an increase in the level of Cyc8p, which is a key repressor of *FLO11* expression and blocks cell adhesion [85]. Unlike Cyc8p, Tup1p upregulates *FLO11* transcription [86]. Additionally, one of the main criteria when choosing yeast strains for primary winemaking is the ability to form hydrogen sulfide, which refers to the hereditary characteristics of the isolate [87]. Exceeding the threshold allowable hydrogen sulfide concentration of 50–80 g/L [87] can adversely affect the taste of wine [88]. Most of the hydrogen sulfide produced during fermentation comes from the biosynthesis of sulfur-containing amino acids (for example, methionine and cysteine).

These amino acids are essential for the growth of *S. cerevisiae*, so yeast strains synthesize them if they are not in the must [89]. So, in a recent work, the authors managed to improve the organoleptic properties of wine by reducing the formation of hydrogen sulfide in an industrial yeast strain, for which two different approaches to improvement were used: hybridization by mass mating and adaptive laboratory evolution [4].

To recommend certain selected promising strains, it is necessary to test them in microwinemaking conditions. Perhaps it will be necessary to adapt the selected strains to higher concentrations of alcohol, which is a necessary condition for the technological regulations in the production of wines of the sherry type [37]. Thus, in one of the works, yeast isolates, selected by genetic markers as "sherry" and showing high oenological indicators on fermented must with an ethanol concentration of 12.5–13.1%, after transferring the film formed by them to wine material with an ethanol concentration of 16%, all died [17]. However, after the gradual adaptation of the selected strains according to the method [90] to an increased concentration of alcohol, they showed a good physiological state, which made it possible to use them for further selection.

## 6. Stress Resistance and Adhesive Characteristics for Screening *S. cerevisiae* Strains

Another promising screening criterion for the selection of flor yeasts is their unique ability for adhesion and response to stress since it is known that during the period of alcoholic fermentation, yeast cells are subjected to various stress factors such as osmotic stress, oxidative stress induced by pro-oxidants such as hydrogen peroxide, high concentration of ethanol, starvation when the supply of nutrients in the environment is depleted, and others [47,99,100]. Osmotic stress is induced by an initially high sugar concentration in grape must (about 20% *w/v*), which the yeast must resist in order for the alcoholic fermentation process to be successfully initiated. As fermentation develops, the concentration of ethanol increases and the load on yeast cells increases, leading to the destruction of their membranes [47,93]. Further yeast cells undergo the process of fortifying with ethanol up to 15.5%, resulting in an increase in the level of acetaldehyde, against the background of a decrease in sugar and nitrogen levels and an increase in oxygen levels [94]. Thus, the strains of *S. cerevisiae* most resistant to abiotic stress factors are promising for sherry production.

In this regard, for an effective search for new candidate strains, a multistage scheme for their screening can be used. At the first stage, genetic screening is carried out [17]; then, the cellular response to short-term exposure to certain toxic agents (ethanol, acetaldehyde, hydrogen peroxide) is studied [95] and the adhesive properties of cells and their ability to form a film are determined [42]. The capacity for sorption on polystyrene and the degree of hydrophobicity of cells according to the nature of their distribution between the aqueous and organic phases can be used as rapid tests for the ability of film formation [91,96,97]. To select strains with cross-resistance to toxic agents, the following criteria are used: ethanol resistance >63%, peroxide resistance >42%, acetaldehyde resistance >50%, high degree of cell hydrophobicity >60%, and sorption on polystyrene $OD_{600} > 1.7$ [42]. Additionally, for the screening of new sherry strains, an analysis of growth characteristics is carried out (duration of the lag phase and exponential growth, cell doubling time, etc.) [92]. Tables S1 and S2 show summary data on screening among sherry, wine and natural isolates of *S. cerevisiae* from the two regions of Eastern Europe, taking into account genetic and oenological markers, stress sensitivity and adhesiveness, with assessment growth curves, which can serve as a guideline for the search for new promising strains for sherry wines.

## 7. Iron Uptake Markers for Screening *S. cerevisiae* Strains

Another important genetic marker that can serve as a selection criterion for sherry strains of *S. cerevisiae* is the polymorphism of the AFT1 and FRE-FIT loci, which makes it possible to select yeast isolates for iron sensitivity [82]. It is known that iron is a vital trace element used in numerous life processes, necessary as a redox co-factor in the transfer of electrons, oxygen, DNA replication and repair, translation, photosynthesis, lipid biosynthesis, nitrogen fixation, etc. [98]. Various organisms have developed complex strategies

to regulate the dynamics of iron concentration in the environment. Thus, in the yeast *S. cerevisiae*, it was shown that iron deficiency leads to the activation of the transcription factor Aft1 and its paralog Aft2, which respond to the mitochondrial signal for the assembly of the Fe-S cluster [99,100]. This entails the activation of more than 30 "Fe-regulon" genes, including metal reductases, high-affinity transporters of free and siderophore-bound iron, intracellular iron transporters, hemoxinases, RNA-binding proteins, etc. [99,101]. On the other hand, an increased concentration of iron has a toxic effect on cells. During the Fenton reaction, when iron ions interact with hydrogen peroxide, hydroxyl radicals and other reactive oxygen species accumulate [102].

To protect cells in response to a high iron content, the transcriptional activator YAP5 protein is activated, upregulating the following genes: *CCC1*, which is the main stimulator of iron accumulation in yeast through its mediated transport into the vacuole; monothiol glutaredoxin *GRX4*, which blocks AFT1/AFT2 transcription factors, reducing their activity; *TYW1*, which encodes an enzyme containing a cytosolic Fe-S cluster, probably involved in iron buffering; and copper metallothionein gene *CUP1* to protect cells from oxidative stress [99,103,104].

In *Aft1p*, the Q648X mutation, specific for flor strains, was found, leading to a truncated form of this protein, without 50 a.a. in the C-terminal domain, but not affecting the glutamine-rich domain. In addition, a 14 kb deletion was found in the right arm of chromosome XV of the studied sherry strains, leading to the elimination of the FRE-FIT cluster [82]. It is known that this cluster encodes the *FRE3* and *FRE5* iron reductase genes, as well as the *FIT2* and *FIT3* genes of GPI-anchored membrane proteins responsible for the retention of siderophore-bound iron in the cell wall during the absence of iron in the medium [105]. It is noteworthy that as a result of the translocation of the terminal region of chromosome XI, a hybrid iron reductase gene *FRE3/FRE2* is formed [73]. At the same time, it is well-known that iron plays an important role in sherry wine, participating in oxidative processes, stimulating the growth and adhesion of yeast cells.

The optimal content of iron in wine materials prepared for sherry is 10–15 mg/L [37,106]. Both decreases and increases in the concentration of iron lead to a slowdown in the formation of the yeast film, a decrease in aroma-forming aldehydes and clouding of the finished product, respectively [37,106]. However, sherry yeast strains of *S. cerevisiae* have historically been selected on wine material from iron-depleted soils [67]. There are iron-sensitive and iron-resistant strains of *Sachoromycetes* [107]. At the same time, there is a correlation between the presence of "sherry" FRE-FIT and AFT1 loci and the degree of sensitivity of screened strains to iron [82]. Iron-sensitive flor strains are better able to accumulate iron at low iron concentrations, while iron-tolerant strains with wine variants of the *AFT1* and *FRE-FIT* better accumulate iron at high concentrations (Supplementary Materials, Table S3). Thus, an additional analysis of the polymorphism of the AFT1 and FRE-FIT loci, which determines the "iron sensitivity" of *S. cerevisiae* cells, can be used in the future as a new useful genetic marker.

## 8. Combined Approach for Screening *S. Cerevisiae* Strains, Promising for Sherry Production

The genetic and oenological markers, as well as the stress response and adhesion parameters, coupled with the ability to grow on materials with low iron content, discussed in Sections 5–7, can be used for multistage screening for new flor yeast strains promising for sherry production among environmental isolates (Figure 4).

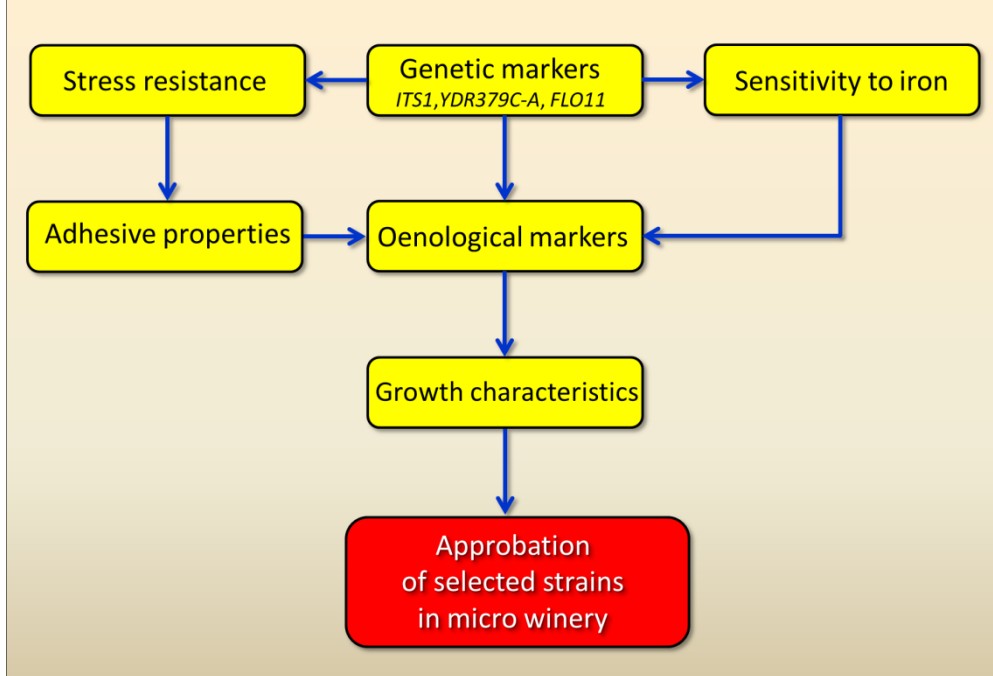

**Figure 4.** Stages of screening yeast *S. cerevisiae*, promising for the production of sherry.

Historically, sherry wines were initially manufactured in Spain in the Andalusia region, where the summer is dry and hot; however, humid air constantly comes in from the Atlantic Ocean, and in addition, the Guadalquivir River provides the region with a mild climate [3,45]. One of the important nutritional features of the vine is moist soil, which is possible due to its composition: chalk, limestone and clay. It is not for nothing that this region is called Albariza, which means "white" [8]. Chalk perfectly absorbs water and then slowly gives it to the roots of the vine, and limestone and clay also retain the moisture well.

However, at present, due to the development of new knowledge in science and technology, the market for wine products is expanding very intensively; in particular, SLW are produced in an increasing number of regions of the world (Figure 1). Successful production of sherry requires specially zoned flor strains adapted for the fermentation of regional grape varieties growing on regional soils. The question of distribution of *S. cerevisiae* in nature remains debatable. Sherry is a specific wine, and the yeast *S. cerevisiae*, which represents a special group of wine-making microflora, plays a significant, if not dominant, role, and it would be impossible to obtain such a wine on an industrial scale [11,12,17]. The occurrence of specific natural strains likely depends on a particular terroir as it is in Spain [108–110]. In addition, it is known that yeast strains can spread efficiently to different regions of the world. For example, it has been shown that the spread of yeast in nature can be facilitated by insects and birds [111–113]; it was also emphasized that migratory birds can only act as carriers of *S. cerevisiae* cells as they can only survive in the gut for 12 h [113]. Additionally, it has been shown that the fruit fly *Drosophila melanogaster*, which has an extremely developed sense of smell and always appears where the fermentation process takes place, can contribute to the spread of yeast [114]. In addition, factors such as raindrops, which wash and carry microorganisms from grapes, and wind, which carries soil and dust particles with microorganisms over long distances, can contribute to the spread of yeast [114]. That is why, in order to improve and develop winemaking in regions with potentially positive nature conditions (geographical latitudes, climate, soils, etc.), it is very important to select new flor yeast strains. This could be absolutely new nature sources, such as orchards, wild grape vines, or cellars of wineries. The type of berries of grape varieties also matters. Thus, *Pedro Ximénez* berries have a thinner skin and more tender juicy pulp than the berries of *Sémillon* and *Aligote* grape varieties with a thick, rough skin

that is more difficult to be damaged by insects and consequently has less yeast appearance on it [114]. In this regard, the complex scheme we have developed (Figure 4) can be a relevant search tool for screening yeast *S. cerevisiae*, promising for the production of sherry in various specific regions of the world.

Microscopic fungi are able to occupy a wide variety of ecological niches in nature. This is possible, among other things, due to their unique metabolic potential, which can be used for a variety of biotechnological processes, from the production of pharmaceutically significant drugs such as antibiotics, statins, immunosuppressants and steroids ([115–118]), to the processing of secondary raw materials [119–121] and the production of enzymes [122–125] and organic acids [126,127]. A special biotechnological role of fungi is in the production of food products, for example, during the fermentation of cheeses [128], wines [129,130] and other food products [131,132]. Modern methods of molecular biology and genetic engineering make it possible to discover the molecular foundations of ongoing processes, with the aim of their further improvement. In the field of obtaining sherry wines, revolutionary events have also taken place recently, including those related to omics approaches [73]. At the molecular level, processes that have been used in traditional winemaking for hundreds and even thousands of years become clear. However, at the same time, understanding the basics of winemaking processes in no way cancels the established technologies; the new knowledge gained can only improve some of these processes in the future [133]. This is why at present it is possible and necessary to apply combined approaches for screening *S. cerevisiae* strains, promising for sherry production.

## 9. Conclusions

The production of sherry wines is a unique, carefully regulated process that uses a special collection of flor-yeast cultures, the storage and maintenance of which is an integral part of the technological process. As a result, specific low-molecular compounds are formed in sherry wines, which determine their unique organoleptic characteristics. Recently, wines using sherry technology have been produced in many countries of the world. In this regard, the challenging task is to search for newly zoned flor strains. In our work, for the first time, an attempt was made to construct a scheme for screening of new promising strains to obtain sherry wines. This appears to be a complex multistage procedure in which it is necessary to take into account a number of key winemaking parameters, which is important for understanding the history of the evolution of flor yeast and for obtaining updated microbial compositions with industrial strains to improve sherry production technology.

**Supplementary Materials:** The following supporting information can be downloaded at: https://www.mdpi.com/article/10.3390/fermentation8080381/s1, Table S1: Genetic, oenological, and growth rate characteristics of *S. cerevisiae* strains; Table S2: Stress resistance, and adhesive characteristics of *S. cerevisiae* strains; Table S3: Average iron concentration in *S. cerevisiae* cells with "sherry" and "wine" variants of the AFT1 and FRE-FIT loci during cultivation on media with different iron concentration.

**Author Contributions:** Conceptualization, D.A.; writing—original draft preparation, D.A.; writing—review and editing, D.A. and A.Z.; visualization, D.A. and A.Z.; supervision, A.Z. All authors have read and agreed to the published version of the manuscript.

**Funding:** This research received no external funding.

**Institutional Review Board Statement:** Not applicable.

**Informed Consent Statement:** Not applicable.

**Data Availability Statement:** Data are contained within the article.

**Conflicts of Interest:** The authors declare no conflict of interest.

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
