# Peer review of "Sherry Wines: Worldwide Production, Chemical Composition and Screening Conception for Flor Yeasts"

_fermentation, doi:10.3390/fermentation8080381_

Round 1
Reviewer 1 Report
Dear Editor and Authors,
The present manuscript focusses on Sherry wines from the regions and the production protocol to the chemical composition and the oenological traits of S. cerevisiae flor yeast. The topic of the review is really interesting and well written. Some expressions should be ameliorated (more details are mentioned below). I also believe that the authors should first talk more general about the consortium of flor microorgansims (including Non-Saccharomyces cerevisiae and bacteria) and then focus on S. cerevisiae which is usually the most representative. Additionally, a figure with the production/aging process will help the reviewer. It could be nice for the conclusion to have more critical points and perspectives as well as expand more the section 8.
More detailed comments:
Line 17
Yeast varieties?? You mean yeast species?
Line 50
Correct durring pls
There are not non- Saccharomyces cerevisiae species that participate? As the title talks about flor yeast in general, I think the authors should make a reference
What about the bacteria?
Table 1.
What do you mean by Sherry like wines (which characteristics of Sherry wines have in common?)
No reference for Cyprus?
Xeres Oreanda (16/)…correct it
Line 88
What do you mean by ‘’deep’’ fermentation and backing technology?
Line 90
Better bag-in-box wines not cask
Line 92
What is the film method??
Line 121
clear gradation?
Line 130
strains of S. cerevisiae or species of Saccharomyces?
Line 264
Which flor yeast do that?
Line 330
cell–cell adhesion? What do you mean?
Line 353
For all yeast flor species?
Line 382
by mass crossing or Mass Mating Hybridization?
Line 471
S. cerevisiae in italics
The last part 8. 8. Combined Approach for Screening S. cerevisiae Strains, Promising for Sherry Production is the most important you should discuss it much more!
Author Response
Response to reviewers
Responses to Managing Editor Ms. Mayora Li
Dear Ms. Mayora Li, thank you very much for the working with our manuscript.
We have made all the required corrections, answered all questions, made the recommended changes and additions to the text of the article.
Responses to Reviewer #1
- The present manuscript focusses on Sherry wines from the regions and the production protocol to the chemical composition and the oenological traits of S. cerevisiae flor yeast. The topic of the review is really interesting and well written. Some expressions should be ameliorated (more details are mentioned below).
RE: We thank the reviewer for the detailed and critical analysis of the manuscript, valuable comments and advices. We have taken into account and made corrections regarding all your comments.
- I also believe that the authors should first talk more general about the consortium of flor microorgansims (including Non-Saccharomyces cerevisiae and bacteria) and then focus on S. cerevisiae which is usually the most representative.
RE: We talk more about the consortium of flor microorgansims (including Non-Saccharomyces cerevisiae and bacteria) in section 3.1, and wine microorganisms were placed in Figure 2. Stage – alcoholic fermentation.
- Additionally, a figure with the production/aging process will help the reviewer.
RE: Thank you very much for this comment. We added figure 2 with the production/aging process of sherry wines.
- It could be nice for the conclusion to have more critical points and perspectives as well as expand more the section 8.
RE: We have expanded the critical points and perspectives in the conclusion section, in accordance with the topics covered in our review and expanded section 8.
More detailed comments:
- Line 17
Yeast varieties?? You mean yeast species?
RE: corrected to flor strains. We would like to clarify, there are many yeast species (for example, C. albicans, C. tropicalis, C. kefyr, C. inconspicua and C. lusitaniae), but only “flor” strains of yeast Saccharomyces cerevisiae play the central role in the sherry manufacturing process.
- Line 50
Correct durring pls
RE: corrected to during
- There are not non- Saccharomyces cerevisiae species that participate? As the title talks about flor yeast in general, I think the authors should make a reference
What about the bacteria?
RE: The role of wine microorganisms (bacteria and non-Saccharomyces yeasts) was further discussed in section 3.1, and wine microorganisms were placed in Figure 2. Stage – alcoholic fermentation.
Table 1.
- What do you mean by Sherry like wines (which characteristics of Sherry wines have in common?)
RE: Sherry-like wines have the same characteristics as sherry wines and are made using the same processes, however they are not eligible to be called sherry wines as they are not registered by the Denominación de Origen (DO) for the production of sherry wines. More about this is written in section 2 - World production of sherry and sherry-like wines (“In addition to Spain, Denominación de Origen (DO) sherry wines have been registered in the European Union in Italy (Sardinia), France (Jura), and Hungary (Tokay Hegyal)… A number of other countries use flor technology as well as deep fermentation and backing technology for sherry-like wines (SLW) production”).
- No reference for Cyprus?
RE: corrected. We added reference for Cyprus.
- Xeres Oreanda (16/)…correct it
RE: corrected
- What do you mean by ‘’deep’’ fermentation and backing technology?
RE: corrected to submerged fermentation and sherry wine aging technology
- Line 90
Better bag-in-box wines not cask
RE: corrected to barrels
- Line 92
What is the film method??
RE: corrected to film-forming method with flor yeasts. Meant - the traditional method for obtaining sherry.
- Line 121
clear gradation?
RE: corrected the whole sentence to “In Spain, the process of making sherry wines of various types has a strict gradation by stages.”
- Line 130
strains of S. cerevisiae or species of Saccharomyces?
RE: strains of S. cerevisiae (flor strains), not species of Saccharomyces. We would like to clarify, there are many yeast species (for example, C. albicans, C. tropicalis, C. kefyr, C. inconspicua and C. lusitaniae), but only “flor” strains of yeast Saccharomyces cerevisiae survive with an increased content of ethanol due to the formation of flor.
- Line 264
Which flor yeast do that?
RE: corrected. We clarified for what flor yeast: “for S. cerevisiae races capensis and bayanus, two typical flor yeasts involved in the biological aging of sherry wines”.
- Line 330
cell–cell adhesion? What do you mean?
RE: we mean, the adhesion of the cell to another cell (adhesion of cells to cells). In a number of works (and in the titles of the works themselves), researchers operate with this term. For example https://www.pnas.org/doi/10.1073/pnas.1801810115
- Line 353
For all yeast flor species?
RE: Yes, this is a characteristic genetic marker for all yeast flor species, it is also necessary to specify that flor strains were found only for S. cerevisiae.
- Line 382
by mass crossing or Mass Mating Hybridization?
RE: Thank you very much for this comment. We corrected to “hybridization by mass mating”
- Line 471
- cerevisiae in italics
RE: corrected
- The last part 8. 8.
Combined Approach for Screening S. cerevisiae Strains, Promising for Sherry Production is the most important you should discuss it much more!
RE: Thanks a lot for your advice. We have expanded the discussion in this section
Responses to Reviewer #2
- Comments and Suggestions for Authors
This review discusses the main sherry-producing regions, chemical composition of sherry wines, as well as genetic, oenological and other selective markers for flor strains that can be used for screening novel candidates promising for sherry production among environmental isolates.
It is interesting and well written. I only have minor remarks.
RE: We are grateful to the reviewer for the careful analysis of the manuscript and valuable comments. We have taken into account and made corrections regarding all your comments.
- Please, improve Fig. 2, adding the metabolic pathways.
RE: Thank you very much for this valuable suggestion.We improve Fig. 2, adding the key metabolic pathways during the fermentation process, covering the formation of such important substances that define sherry wines as glycerol, acetaldehyde and ethanol. However, compounds in sherry species are formed as a result of thousands of metabolic reactions, so we could not fit them all in the diagram (they would be too small and unreadable). Therefore, we concentrated on the basic processes of alcoholic fermentation for sherry wines production.
- Par. 3: please prepare a table to summarize the manufacturing processes.
RE: In accordance with your important remark, we have prepared Table 2 in which we summarize the manufacturing processes of sherry wines. To facilitate understanding, we have also introduced an additional figure 2, which graphically displays the stages of obtaining sherry wines. Table 2 and Figure 3 are now in section 3.
- Par5, 6, 7: S. cerevisiaein italics
Please, write in italics all genes
RE: corrected

Reviewer 2 Report
This review discusses the main sherry-producing regions, chemical composition of sherry wines, as well as genetic, oenological and other selective markers for flor strains that can be used for screening novel candidates promising for sherry production among environmental isolates.
It is interesting and well written. I only have minor remarks. Please, improve Fig. 2, adding the metabolic pathways.
Par. 3: please prepare a table to summarize the manufacturing processes.
Par5, 6, 7: S. cerevisiae in italics
Please, write in italics all genes
Author Response
Responses to Reviewer #2
- Comments and Suggestions for Authors
This review discusses the main sherry-producing regions, chemical composition of sherry wines, as well as genetic, oenological and other selective markers for flor strains that can be used for screening novel candidates promising for sherry production among environmental isolates.
It is interesting and well written. I only have minor remarks.
RE: We are grateful to the reviewer for the careful analysis of the manuscript and valuable comments. We have taken into account and made corrections regarding all your comments.
- Please, improve Fig. 2, adding the metabolic pathways.
RE: Thank you very much for this valuable suggestion.We improve Fig. 2, adding the key metabolic pathways during the fermentation process, covering the formation of such important substances that define sherry wines as glycerol, acetaldehyde and ethanol. However, compounds in sherry species are formed as a result of thousands of metabolic reactions, so we could not fit them all in the diagram (they would be too small and unreadable). Therefore, we concentrated on the basic processes of alcoholic fermentation for sherry wines production.
- Par. 3: please prepare a table to summarize the manufacturing processes.
RE: In accordance with your important remark, we have prepared Table 2 in which we summarize the manufacturing processes of sherry wines. To facilitate understanding, we have also introduced an additional figure 2, which graphically displays the stages of obtaining sherry wines. Table 2 and Figure 3 are now in section 3.
- Par5, 6, 7: S. cerevisiaein italics
Please, write in italics all genes
RE: corrected
